# The Role of *Bifidobacterium* in Liver Diseases: A Systematic Review of Next-Generation Sequencing Studies

**DOI:** 10.3390/microorganisms11122999

**Published:** 2023-12-17

**Authors:** Gabriel Henrique Hizo, Pabulo Henrique Rampelotto

**Affiliations:** 1Graduate Program in Gastroenterology and Hepatology Sciences, Universidade Federal do Rio Grande do Sul, Porto Alegre 90035-003, Brazil; 2Bioinformatics and Biostatistics Core Facility, Instituto de Ciências Básicas da Saúde, Universidade Federal do Rio Grande do Sul, Porto Alegre 91501-907, Brazil

**Keywords:** microbiome, microbiota, amplicon sequencing, non-alcoholic fatty liver disease, alcoholic liver disease, hepatocellular carcinoma

## Abstract

The physiopathology of liver diseases is complex and can be caused by various factors. *Bifidobacterium* is a bacterial genus commonly found in the human gut microbiome and has been shown to influence the development of different stages of liver diseases significantly. This study investigated the relationship between the *Bifidobacterium* genus and liver injury. In this work, we performed a systematic review in major databases using the key terms “Bifidobacterium”, “ALD”, “NAFLD”, “NASH”, “cirrhosis”, and “HCC” to achieve our purpose. In total, 31 articles were selected for analysis. In particular, we focused on studies that used next-generation sequencing (NGS) technologies. The studies focused on assessing *Bifidobacterium* levels in the diseases and interventional aimed at examining the therapeutic potential of *Bifidobacterium* in the mentioned conditions. Overall, the abundance of *Bifidobacterium* was reduced in hepatic pathologies. Low levels of *Bifidobacterium* were associated with harmful biochemical and physiological parameters, as well as an adverse clinical outcome. However, interventional studies using different drugs and treatments were able to increase the abundance of the genus and improve clinical outcomes. These results strongly support the hypothesis that changes in the abundance of *Bifidobacterium* significantly influence both the pathophysiology of hepatic diseases and the related clinical outcomes. In addition, our critical assessment of the NGS methods and related statistical analyses employed in each study highlights concerns with the methods used to define the differential abundance of *Bifidobacterium*, including potential biases and the omission of relevant information.

## 1. Introduction

The study of the microbiota in liver diseases has gained significant interest due to the emerging understanding of the intricate relationship between gut microbial communities and liver health. Studies have investigated the significant function of the gut–liver axis as a feedback loop in the gastrointestinal system. It is thought that alterations in the gut microbiota are closely associated with an increase in intestinal permeability. This alteration can lead to the release of endotoxins, triggering inflammatory responses and affecting bile acid metabolism [1]. Accumulating evidence suggests that the gut–liver axis plays a pivotal role in various liver conditions, from Alcoholic Liver Disease (ALD) and Non-alcoholic Fatty Liver Disease (NAFLD) to Hepatocellular Carcinoma (HCC) [2,3]. Dysbiosis, a term used to describe the imbalance of the gut microbiota, has been linked to inflammation, metabolic dysfunction, and the progression of liver diseases [4]. Investigating the composition and function of the microbiota in these contexts holds promise for identifying novel biomarkers, therapeutic targets, and interventions that could revolutionize the management and prevention of liver diseases.

One of the key members of the microbiota in liver diseases is the genus *Bifidobacterium*. *Bifidobacterium* is an essential genus of bacteria found in the gastrointestinal tract. Its main protective mechanisms include improved adherence to the intestinal barrier, increased production of short-chain fatty acid (SCFA) and pH reduction, the release of bifidocins (peptides that inhibit the growth of other pathogenic bacteria), and positive immunomodulatory effects [5]. Nevertheless, the levels of *Bifidobacterium* are often reduced in liver injuries, which may be directly linked to the pathophysiology of the disease. Consequently, recent studies have focused on how probiotics can be used to supplement this deficiency and potentially treat liver diseases more effectively [6].

Next-generation sequencing (NGS), also known as high-throughput DNA sequencing, has become the preferred method for analyzing microbiotas in various human pathologies, including liver diseases. The most common approach to identifying bacteria based on these modern technologies is through amplicon sequencing or shotgun metagenomics. Amplicon sequencing involves PCR amplification and sequencing of the 16S ribosomal RNA (16S rRNA) gene [7]. This technique is relatively cost-effective and provides a targeted analysis of the microbial taxa in each sample, providing valuable insights into the composition and diversity of the microbiota associated with different clinical conditions [8]. However, shotgun metagenomics is an alternative approach in which the entire microbial DNA is sequenced without any prior amplification or targeting. This method provides a more comprehensive view of the microbial community, including bacteria, viruses, fungi, and other microorganisms [9]. Shotgun metagenomics allows for the identification of novel pathogens and the exploration of functional potentials within the microbiota [10]. However, it can be more expensive and computationally intensive compared to amplicon sequencing.

For nearly a decade, NGS technologies have been used to study the bacterial communities in liver diseases [11]. These studies have employed different sampling techniques, sequencing technologies, and statistical methods. The diversity of these factors makes it challenging to obtain a comprehensive overview of the field and identify the main trends in the microbial composition associated with each stage of the disease, especially when it comes to scientific but relevant taxa such as the genus *Bifidobacterium*.

In this systematic review, we conducted a comprehensive and critical review of research that has explored the relationship between the *Bifidobacterium* genus and liver diseases, from NAFLD and ALD to HCC. We specifically focused on studies that used next-generation sequencing (NGS) technologies to investigate this association, providing, for the first time, an in-depth analysis of the methods in liver diseases and the identification of this particular genus. Our goal was to analyze and combine the results of studies that explore the potential impact of *Bifidobacterium* on liver damage and its possible correlation with disease advancement. We conducted a thorough review of existing literature to achieve this objective. By studying how *Bifidobacterium* exerts these effects, we can gain a better understanding of the underlying mechanisms of liver injuries and potentially develop new treatments that target these mechanisms.

## 2. Materials and Methods

### 2.1. Search Strategy and Selection Criteria

A systematic literature search was conducted using the PubMed, Embase, and Web of Science databases. The investigation was limited to articles published in English from January 2013 to June 2023. The search terms included “Bifidobacterium”, “NAFLD”, “NASH”, “ALD”, “cirrhosis”, and “HCC”, along with their medical subject headings (MeSH in PubMed and Emtree in Embase). The search strategy was adapted and standardized across all databases with careful application of logical connectors. The Appendix A contains a detailed search strategy for each database. Studies that did not meet these criteria were excluded, excluding reviews, case reports, editorials, book chapters, conference abstracts, and notes from the outset. In particular, we focused on studies that used next-generation sequencing (NGS) technologies to investigate the potential impact of *Bifidobacterium* on liver damage. Six hundred forty-eight initial studies were identified through a customized strategy for systematic database search, as shown in Figure 1.

The Rayyan web platform was used to identify and remove duplicate reports [12]. Initially, article relevance was evaluated based on their titles and abstracts, and a comprehensive content analysis was conducted when necessary. The selected papers then underwent a thorough assessment, with relevant data extracted for research analysis. After removing duplicates and studies that did not fit the scope of this review, a total of 31 studies were selected to form the representative sample. Subsequently, these studies were organized according to their respective categories: observational and observational with treatment. The Preferred Reporting Items for Systematic Reviews and Meta-Analyses (PRISMA) criteria were carefully adhered to in this review, as shown in Appendix A [13].

### 2.2. Data Extraction and Analysis

The data from the selected articles were organized into a dataset containing the following variables: authors’ names, publication year, country, disease, population (human or animal), type of study (epidemiological classification), sample type, next-generation sequencing technology, sequencing type, sample size (N), groups, statistical methodology, database, and clinical outcomes. These data are available in Appendix A.

Risk-of-bias analysis was conducted using validated approaches from the Joanna Briggs Institute (JBI) to assess the quality of studies [14]. JBI tools provide robust criteria and standards to guide study quality.

## 3. Results

### 3.1. Selected Studies

Table 1 presents the results of 17 studies that evaluated the levels of *Bifidobacterium* in both control and disease groups, encompassing human and animal samples. These findings were developed in various countries, with China leading the number of publications.

In Table 2, although the studies presented disease groups separately, the *Bifidobacterium* levels for those groups were not measured. Only the disease group with treatment had available abundance values for *Bifidobacterium*.

In general, the 31 studies used two types of sequencing: 16S rRNA amplicon sequencing and shotgun metagenomics (Figure 2A). In 16S rRNA sequencing, the most prevalent region was V3–V4, followed by V4. Regions V1–V3 and V4–V5 appeared less frequently. There was considerable variability in the databases used for taxonomic classification (Figure 2A). The most commonly used databases for 16S rRNA classification were SILVA and Greengenes, while MetaPhIAn was used for shotgun metagenomic analysis. It is noteworthy that several studies did not report the databases used. Concerning the statistical analysis used to identify differentially abundant genera between control and disease groups, Linear Discriminant Analysis of Effect Size (LEfSe) and Mann–Whitney were the most employed methods (Figure 2A). However, many studies also resorted to simple statistical tests, such as Mann–Whitney, Wilcoxon, Kruskal–Wallis, and Student’s *t*-test. Once again, many studies did not report the statistical methods used for abundance differentiation. These findings highlight the diversity of methodological approaches used in *Bifidobacterium* studies and the need for greater transparency in disclosing the methods employed to strengthen the reliability and comparability of the results obtained.

The levels of *Bifidobacterium* in liver diseases were assessed in several observational studies. Out of 17 studies, only four reported elevated levels of *Bifidobacterium* in the disease group compared to the control group (Figure 2B). Two of these studies involved NAFLD and the other two involved cirrhosis. In contrast, all other studies reported reduced levels of *Bifidobacterium* in the disease groups compared to control.

In addition, 14 studies observed *Bifidobacterium* levels in the disease groups after receiving some form of treatment, such as a drug, molecule, or probiotic that did not contain *Bifidobacterium* in its composition (Figure 2C). All these studies demonstrated an increase in *Bifidobacterium* levels after treatment.

### 3.2. Risk-of-Bias and Quality Assessment

Out of the 31 studies reviewed, 17 followed a clinical model and were categorized by their epidemiological approaches: case-control, cross-sectional, and cohort. The case-control and cross-sectional studies had a high risk of bias levels, while the cohort studies exhibited a low risk of bias. All information regarding bias risk is displayed in Figure 3.

## 4. Discussion

This review highlights the strong relationship between the *Bifidobacterium* genus and different hepatic lesions (including ALD, NAFLD, NASH, cirrhosis, and HCC). Remarkably, most studies have revealed significantly reduced levels of *Bifidobacterium* in these hepatic conditions compared to control groups. Additionally, all studies that employed treatments for these diseases observed restoring *Bifidobacterium* levels. These results strongly support the hypothesis that this bacterial genus plays a pivotal role in the health status of humans as a member of the beneficial microbiota, and major changes in its abundance significantly influence both the pathophysiology of hepatic diseases and their therapeutic approach. Furthermore, we thoroughly evaluated the NGS methodologies and associated statistical analyses employed in each study. We specifically examined the methods used to determine the differential abundance of *Bifidobacterium*, highlighting concerns such as potential biases and the omission of relevant information in certain studies.

The discussion section was organized to comprehensively understand the relationship between *Bifidobacterium* and the pathophysiology of each liver injury. Initially, the section presented the role of *Bifidobacterium* in maintaining the balance of the intestinal microbiota and its contribution to a healthy phenotype. Subsequently, the relationship between *Bifidobacterium* and different liver conditions was addressed, covering the progression and stages of liver diseases in Section 4.1, the *Bifidobacterium* genus in Section 4.2, ALD in Section 4.3, NAFLD in Section 4.4, NASH in Section 4.5, cirrhosis in Section 4.6, and HCC in Section 4.7. Each section explored the underlying mechanisms of action and the potential therapeutic effects of *Bifidobacterium* for each liver condition. This approach aims to provide a comprehensive understanding of the influence of *Bifidobacterium* on each liver injury and its potential clinical implications.

### 4.1. The Progression and Stages of Liver Diseases

The liver plays a crucial role in digestion and metabolism, making it vulnerable to the effects of diet and alcohol consumption. Although the physiological processes share common mechanisms, it is possible to identify a non-alcoholic pathway, more closely related to the excessive accumulation of fat, and an alcoholic pathway, triggered by the toxicity of excessive alcohol consumption (Figure 4). The non-alcoholic pathway progresses through the stages of NAFLD and NASH, ultimately leading to extensive fibrous tissue and the characteristic inflammatory process of a cirrhotic liver [45].

Similar to NAFLD, the alcoholic pathway begins with the accumulation of fat in liver cells due to excessive alcohol consumption. This condition is reversible if alcohol consumption is stopped. Prolonged and heavy alcohol consumption can lead to inflammation and liver cell damage. This stage is known as alcoholic steatohepatitis. Some individuals with alcoholic steatohepatitis can experience rapid disease progression. Advanced fibrosis can progress to cirrhosis in ALD as well [46]. In both pathways, the severity of cirrhosis can lead to HCC, the most advanced stage of liver cancer, as well as a loss of liver function and, often, death [47].

### 4.2. The Bifidobacterium Genus

In 1900, Henri Tissier discovered a rod-shaped bacterium initially thought to belong to the *Lactobacteriaceae* family. This bacterium was gram-positive, catalase-negative, anaerobic, non-spore-forming, non-gas-producing, and non-motile. After new research emerged, it was determined that *Bifidobacterium* is a genus that pertains to the Actinobacteria phylum, thereby distinguishing bifidobacteria from lactobacilli [48]. According to the List of Prokaryotic Names with Standing in Nomenclature (LPSN), about 90 species have been identified until now, with some of the most-studied and well-known being *B. adolescentis, B. infantis, B. bifidum, B. breve,* and *B. longum* [49].

*Bifidobacterium* constitutes a relevant part of the human body’s microbiota and is present in various regions of the organism, such as the oral cavity, gastrointestinal tract, vagina, and cervix [50]. At the time of birth, due to fluid exchange between mother and fetus, the levels of *Bifidobacterium* are higher. As the child develops and transitions to adulthood, there is a separation from maternal feeding and consequent ingestion of nutritionally poor foods, resulting in a drastic reduction in *Bifidobacterium* levels [51].

The presence of the *Bifidobacterium* genus in the composition of the human gastrointestinal microbiota has been associated with numerous health benefits. For instance, in the human diet, undigested dietary fibers are fermented by *Bifidobacterium*. In this process, SCFAs are produced, which, in turn, are associated with intestinal cell health and immune system regulation [52]. Moreover, the mere presence of *Bifidobacterium* helps maintain the microbiota balance as they compete for space and nutrients, limiting the growth of potentially harmful bacteria. Consequently, this leads to positive immune system modulation, intestinal barrier strengthening, and nutrient absorption improvement [53].

Additionally, studies have shown that levels of *Bifidobacterium* are reduced in various conditions such as diabetes and major depression [54,55]. Some analyses explore the potential of *Bifidobacterium* levels in aiding disease diagnosis and treatment [56].

### 4.3. Bifidobacterium in ALD

ALD is the result of chronic and excessive alcohol consumption. Although there is no well-defined minimum threshold, it is known that ingesting around 40 g of pure alcohol per day over many years significantly increases the risk of developing ALD [3]. ALD is a condition that varies in severity, ranging from early stages of hepatic steatosis to more severe complications, such as cirrhosis and HCC [57].

Changes in gut microbiota associated with ALD include increased gram-positive bacteria, such as the *Enterobacteriaceae* family, linked to a pro-inflammatory state. However, bacteria from the Clostridia class, which produce SCFAs like butyrate, are less abundant [20].

This review included only one study according to the established criteria. As expected, *Bifidobacterium* levels were reduced in the disease group, which correlated with a pronounced pro-inflammatory profile [15]. The decrease in *Bifidobacterium* in ALD is caused by alcohol toxicity, which disrupts gut mucosa and microbiota [58].

The hypothesis that reduced levels of *Bifidobacterium* in ALD are related to metabolic and microbiota imbalances gains more strength when observing the results of studies using treatments for the disease. In animal models, it was observed that the diseased group receiving human beta-defensin 2 (hBD-2) showed an increase in *Bifidobacterium* abundance, which may be associated with less hepatic fat accumulation, reduced hepatocellular injury, and inflammation [31]. In another animal study, treatment with auricularia auricula melanin (AMM) also elevated *Bifidobacterium* levels in the diseased group, along with reducing harmful bacteria presence and improving intestinal mucosal health [32]. In a clinical trial, ALD patients who received fecal microbiota transplantation also experienced increased levels of *Bifidobacterium* and improvements in digestive and immune health due to microbial modulation [33].

In conclusion, chronic alcohol consumption disrupts the gut microbiota, reducing the abundance of the beneficial genus *Bifidobacterium* and increasing pro-inflammatory bacteria. Studies show that treatments promoting *Bifidobacterium* increase can improve liver health and reduce inflammation in ALD.

### 4.4. Bifidobacterium in NAFLD

NAFLD is a hepatometabolic disease characterized by at least 5% steatosis in hepatocytes caused by abnormal fat accumulation. However, in the development of NAFLD, the fatty accumulation is not caused by excessive alcohol consumption or other autoimmune metabolic disorders, although the presence of comorbidities such as Type 2 diabetes and hypertension is familiar [59]. The presence of fat and the resulting metabolic alterations directly affect the loss of diversity and modification of the gastrointestinal microbiota composition [60].

This microbiota imbalance is related to the dysregulation in the production of SCFA and the inhibition of fasting-induced adiposity factor (FIAF) secretion. These changes lead to a biochemical cascade that induces fat accumulation in adipocytes and hepatocytes through lipogenesis. Additionally, dysbiosis is involved in increased lipopolysaccharide (LPS) production, which stimulates a pro-inflammatory response through Toll-like receptor 4 (TLR4) activation and promotes Kupffer cell activation. Interestingly, although NAFLD is not caused by excessive alcohol consumption, the bacteria, in the context of dysbiosis, can produce ethanol endogenously. This phenomenon increases intestinal permeability, toxins, and inflammation while reducing nutrient absorption [61].

Despite most studies reviewed in this article corroborating with the literature and pointing to a reduction in *Bifidobacterium* levels in NAFLD patients, two exceptional studies reported an increase in the levels of this bacterium. It is important to note that, in such exceptional cases, the increased levels of *Bifidobacterium* were not associated with harmful effects. This finding challenges the notion that any deviation in *Bifidobacterium* levels is detrimental, indicating a need for further research to comprehend the adaptive mechanisms of the microbiota in the context of NAFLD.

The discrepancies observed among studies regarding the levels of *Bifidobacterium* in NAFLD patients may be attributed to different factors. Firstly, the composition of the gut microbiota is highly individualized and can vary significantly between individuals [62]. This natural variation in gut microbiota composition can contribute to differences in *Bifidobacterium* levels observed among different study populations. Second, NAFLD itself is a complex condition with various underlying causes and disease progression patterns. Factors such as diet, lifestyle, genetics, and co-existing health conditions can influence the composition of the gut microbiota and its relationship with NAFLD [63]. The phenomenon of the adaptive response of the microbiota demonstrates its dynamic capacity to adapt to the environment and the inflammatory mechanisms of the human body [64]. Thus, variations in these factors among study populations may also contribute to the observed discrepancies. In addition, differences in the timing of sample collection and the duration of the study period may also contribute to the significant differences in results [65]. Lastly, the methods used to analyze and measure *Bifidobacterium* levels can vary between studies. Differences in sequencing techniques and data analysis can all influence the accuracy and comparability of the results. Therefore, inconsistencies in the methodologies employed by different research groups can contribute to the discrepancies observed.

Studies observing groups with NAFLD who received treatment have consistently shown increased *Bifidobacterium* levels. This supports the hypothesis that the disease’s development is linked to an imbalance in gut bacteria, specifically a decrease in *Bifidobacterium*. To achieve the best possible clinical results, it is essential to concentrate on restoring the levels of *Bifidobacterium*. Inulin, for example, has been a potential treatment for NAFLD, which has gained prominence in studies. It is a critical dietary fiber in the production of SCFA, stimulating the growth of *Bifidobacterium*. Consequently, there is an improvement in intestinal barrier health and the immune system [66].

In summary, NAFLD is linked to an imbalance in gut microbiota, specifically a decrease in *Bifidobacterium* levels and altered SCFA production. A potential approach to treating NAFLD is restoring *Bifidobacterium* levels through interventions like inulin. Studies on animals and humans have shown that certain *Bifidobacterium* species have inhibitory effects on NAFLD, indicating that gut microbiota is a promising target for therapy.

### 4.5. Bifidobacterium in NASH

NASH is a condition that is part of the NAFLD spectrum but is considered more severe due to the progression of inflammation and the development of liver lesions [67]. NAFLD is characterized by the accumulation of fat in the liver, which can evolve into NASH. In the latter, inflammation levels are typically higher, and intestinal permeability becomes more impaired, leading to a more significant imbalance in the microbiota [59].

Similar to NAFLD, in NASH, there is an increase in the genera *Provatella* and *Escherichia*, which are gram-negative bacteria associated, for example, with increased ethanol production [68]. However, the genus *Bifidobacterium* shows a decrease in abundance as the disease progresses [2]. All studies included in this review that evaluated *Bifidobacterium* levels in NASH confirmed these findings. This reduction in *Bifidobacterium* can exacerbate the microbiota imbalance, worsening intestinal permeability and directly impacting liver lipid accumulation, insulin resistance, and fibrosis development [59].

Observational studies on treatments included in this review support the idea that reducing *Bifidobacterium* levels harms the patient’s biochemical and clinical status. For example, after liver transplantation in NASH patients, an increase in *Bifidobacterium* levels was observed, possibly correlated with reduced hepatic fat [43]. Patients who received oligofructose also had their *Bifidobacterium* abundance restored, which was accompanied by reduced inflammation and fibrosis [44].

These findings emphasize the importance of the intestinal microbiota in NASH progression and suggest that strategies to preserve or restore *Bifidobacterium* levels may be beneficial in treating this liver condition.

### 4.6. Bifidobacterium in Cirrhosis

Cirrhosis is a clinical condition characterized by fibrosis resulting from recurrent wound-healing events in the hepatic tissue. These wounds are, in turn, a consequence of excessive fat accumulation and chronic alcohol exposure [69]. This context also promotes increased vascularization in the liver and portal hypertension [70]. Not surprisingly, liver transplantation remains the best curative option, although some drugs are already under study and show therapeutic potential [71].

Due to these pathophysiological mechanisms, cirrhosis is a disease intimately affected by the dysregulation of the gut microbiota. The simultaneous increase in *Enterobacteriaceae*, *Enterococcaceae*, and *Streptococcaceae* families, as well as the decrease in *Lachnospiraceae* and *Ruminococcaceae* families, strongly suggests elevated toxin production, decreased conversion of primary bile acids into secondary bile acids, and reduced production of SCFA [72].

In this review, the included observational studies did not provide conclusive results regarding the levels of *Bifidobacterium* in cirrhotic patients. One study reported a lower abundance of *Bifidobacterium* in patients, related to worsened inflammatory profile and liver damage [18]. On the other hand, two other studies found higher levels of *Bifidobacterium* in disease groups compared to control groups. Although it was a different result than expected, the researchers discuss that similar results have been found in other diseases, mainly cardiac and renal diseases. However, there is no correlation between a higher presence of *Bifidobacterium* and worse clinical outcomes in cirrhotic patients [19,20].

Although there is no clear definition of whether *Bifidobacterium* levels are reduced or increased in cirrhotic patients, treatment for the disease can elevate these levels. Furthermore, it appears that the cirrhotic patient’s health improves with increasing levels of *Bifidobacterium*. A recent study delved into the therapeutic effects of lactitol. The patients who were administered this treatment showed an abundance of *Bifidobacterium* and experienced improved health. The treatment also reduced the biosynthesis of endotoxin and inflammatory components [35].

Therefore, while studies confirmed that the intestinal microbiota affects cirrhosis, it is not established how *Bifidobacterium* changes in cirrhotic patients. More research is needed to better understand its role and therapeutic potential.

### 4.7. Bifidobacterium in HCC

HCC is the most common form of liver cancer in adults and is one of the leading causes of global mortality. In recent years, the predominant cause of HCC development has shifted from primarily being related to viral hepatitis to being associated with increased alcohol consumption and liver fat accumulation. Both patients with ALD and those with cirrhosis have a high risk of progression to HCC [73].

Excess alcohol and liver fat accumulation cause metabolic alterations that, over time, lead to crucial genomic changes in hepatocytes. The significant alterations involve genetic mutations, such as in the Telomerase Reverse Transcriptase (TERT) promoter and Tumor Protein 53 (TP53) gene, epigenetic modifications, and changes in growth factors, such as the Wnt/β-catenin and tyrosine kinase pathways [74].

This complex pathogenesis, with multiple interconnected mechanisms, also significantly affects the microbiota in HCC. Various imbalances are found, including the enrichment of bacteria such as *Veillonella*, *Streptococcus*, *Clostridium*, and *Provotella*, as well as the reduction of beneficial bacteria such as *Eubacterium*, *Faecalibacterium*, and *Bifidobacterium* [75].

Results from studies consistently demonstrated that both in animal models and clinical trials, *Bifidobacterium* levels are reduced in HCC [3,47,48]. An imbalanced microbiota, especially with lower levels of *Bifidobacterium*, can trigger a series of events, such as increased pathogenic bacteria, higher production of toxic and pro-carcinogenic substances, chronic inflammation, increased oxidative stress, DNA damage, progression of liver fibrosis, and cell death.

However, treatments with Tremelimumab and Durvalumab, two immunotherapeutic drugs, have shown clinical remission in HCC patients, and this effect may be related to the increase in *Bifidobacterium* levels observed after treatment. Beneficial bacteria such as *Bifidobacterium* and *Akkermansia* may enhance treatment response, mainly due to their anti-inflammatory activity and production of SCFA [34].

These findings highlight the importance of intestinal microbiota in HCC pathogenesis and treatment. They may provide insights for developing new therapeutic approaches based on the manipulation of *Bifidobacterium* to improve clinical outcomes for these patients.

### 4.8. Risk-of-Bias and Quality Assessment

Only three studies identified confounding factors. Huo, R. et al. reported that patients with advanced HCC received various treatments [4]. Wei, X. et al. also emphasized that using medications, diverse diets, and comorbidities could introduce bias [19]. Duarte et al. mentioned that the NASH group was composed only of female patients, which could have influenced the results [29]. The other studies did not report if there were any confounding factors, which is concerning since this is a common occurrence in clinical settings.

The methods used to detect and measure the *Bifidobacterium* genus in the microbiome can also offer insights into the reliability of the research papers. In particular, the NGS technology, the sequencing type, the reference database, and the statistical methods used to define the differential abundance of the genus among groups are of primary relevance.

Several studies on amplicon sequencing use the V3–V4 region, which is the standard in microbiome studies. However, some papers use other 16S regions, such as V3, V4–V5, and V5–V6, making it difficult to compare results across studies. The database is a significant concern in several studies, mainly because they rely on Greengenes, an ancient and outdated database last updated in 2013. On the other hand, the SILVA and the Ribosomal Database Project (RDP) continuously receive taxonomy revisions and updates. Furthermore, six studies did not identify which database they used, which is detrimental to research and its reproducibility.

Another important point concerns the statistical method used to assess the differences in bacterial abundance among different groups. Some studies used advanced methods such as LEfSe for microbiome analysis. In contrast, others relied on more straightforward statistics, such as the Krustal–Wallis or Wilcoxon test. The statistical methods used to analyze the microbiota need to be well-considered. A recent study demonstrated that using the Wilcoxon test on normalized data with a centered log-ratio led to the detection of up to 90% of bacteria in two sample groups. However, when employing Analysis of Compositions of Microbiomes (ANCOM)—a more robust, current, and suitable mathematical tool for microbiota analysis—bacterial detection decreased to 0.8% in the same sample data [76]. Other analytically more effective microbiome techniques than classical methods include LEfSe and Analysis of Compositions of Microbiomes with Bias Correction (ANCOM–BC). Among these, ANCOM–BC, being the most contemporary, offers additional advantages such as sampling fraction control, integrated statistical tests, confidence intervals, and considerable computational efficiency [77]. However, no study used the ANCOM–BC method. Furthermore, six studies omitted the statistical method utilized to assess variances in the *Bifidobacterium* genus, which is problematic.

It is necessary to reduce bias to ensure the quality of study results. This can be achieved through proper design and following validated checklists such as JBI. Additionally, choosing an updated database and using robust statistical methodologies for microbiota analysis should be considered to better understand the relationship between the *Bifidobacterium* genus and liver disease. On this aspect, new checklists have been proposed to improve reporting in future human microbiome studies, such as the “Strengthening The Organization and Reporting of Microbiome Studies” (STORMS) [78].

## 5. Conclusions

This comprehensive review has revealed a reduced abundance of *Bifidobacterium* in all stages of liver disease, such as ALD, NAFLD, NASH, cirrhosis, and HCC. However, interventional studies using different drugs and treatments were able to increase the abundance of the genus and improve clinical outcomes. By evaluating, in detail, the different studies, this review sheds light on the complex relationship between *Bifidobacterium* and liver diseases, as well as its progression in different stages. Various mechanisms are at play in this process, including the amplification of SCFA production, the dampening of inflammation, and the reinforcement of intestinal mucosal integrity. In addition, we have critically accessed the NGS methods and related statistical analyses of each study, highlighting concerns with the methods used to define the differential abundance of *Bifidobacterium* in some studies, including potential bias and the omission of relevant information.

Our better understanding of the underlying mechanisms of how this particular genus is related to liver injuries may help the development of new treatments that target these mechanisms. Furthermore, choosing an updated database and using robust statistical methodologies for microbiota analysis should be considered to better understand the relationship between the *Bifidobacterium* genus and liver disease.

## Figures and Tables

**Figure 1 microorganisms-11-02999-f001:**
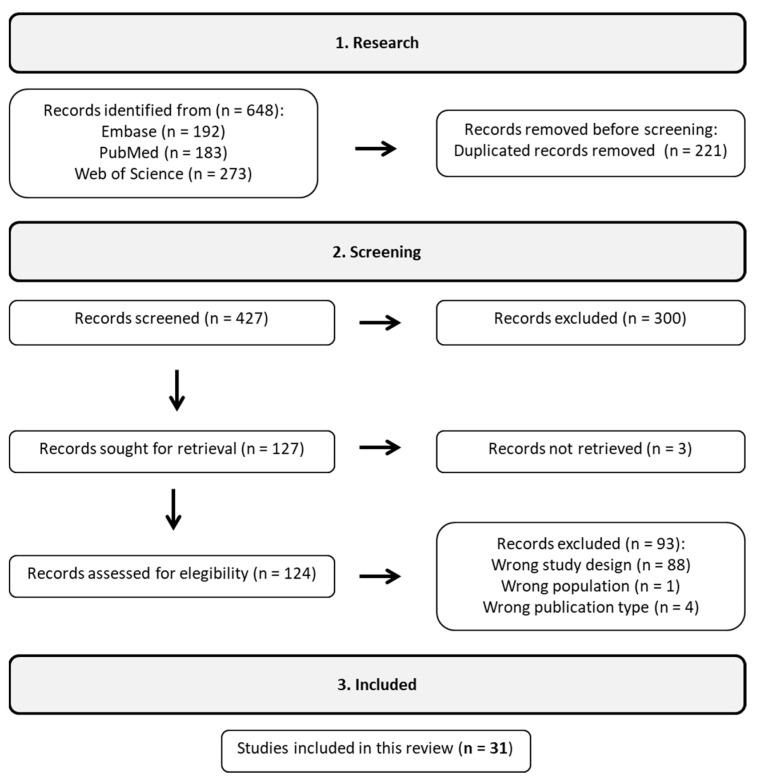
According to the workflow, 31 studies were included out of the initial 648 following a rigorous filtering process.

**Figure 2 microorganisms-11-02999-f002:**
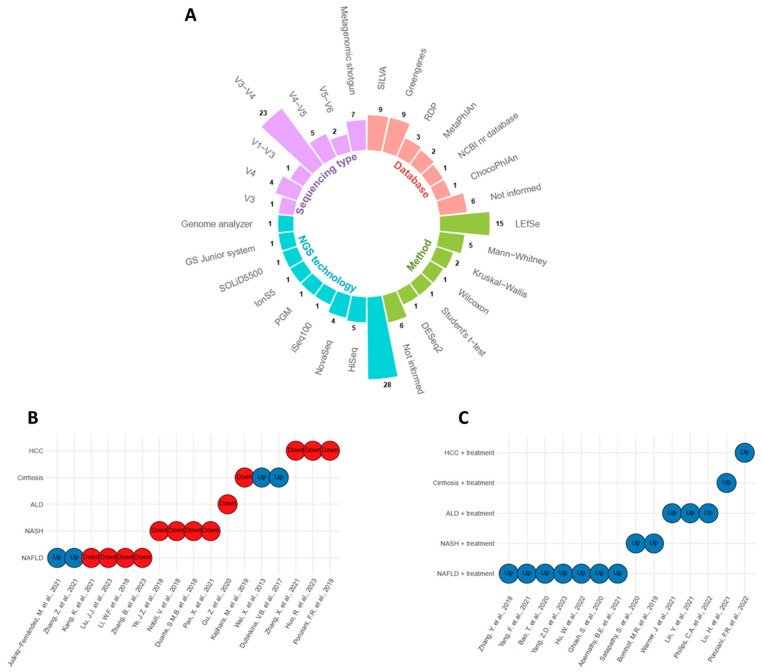
Overview of the methods used to detect and quantify *Bifidobacterium* genus in liver lesions and their upregulation or downregulation status. (**A**) Circular barplot with sequencing type, database, method, and next-generation sequencing (NGS) technology information. (**B**) *Bifidobacterium* levels in untreated disease groups [4,15,16,17,18,19,20,21,22,23,24,25,26,27,28,29,30], see Table 1. (**C**) *Bifidobacterium* levels in treated disease groups [31,32,33,34,35,36,37,38,39,40,41,42,43,44], see Table 2. Blue circles denote upregulation, and red circles indicate downregulation.

**Figure 3 microorganisms-11-02999-f003:**
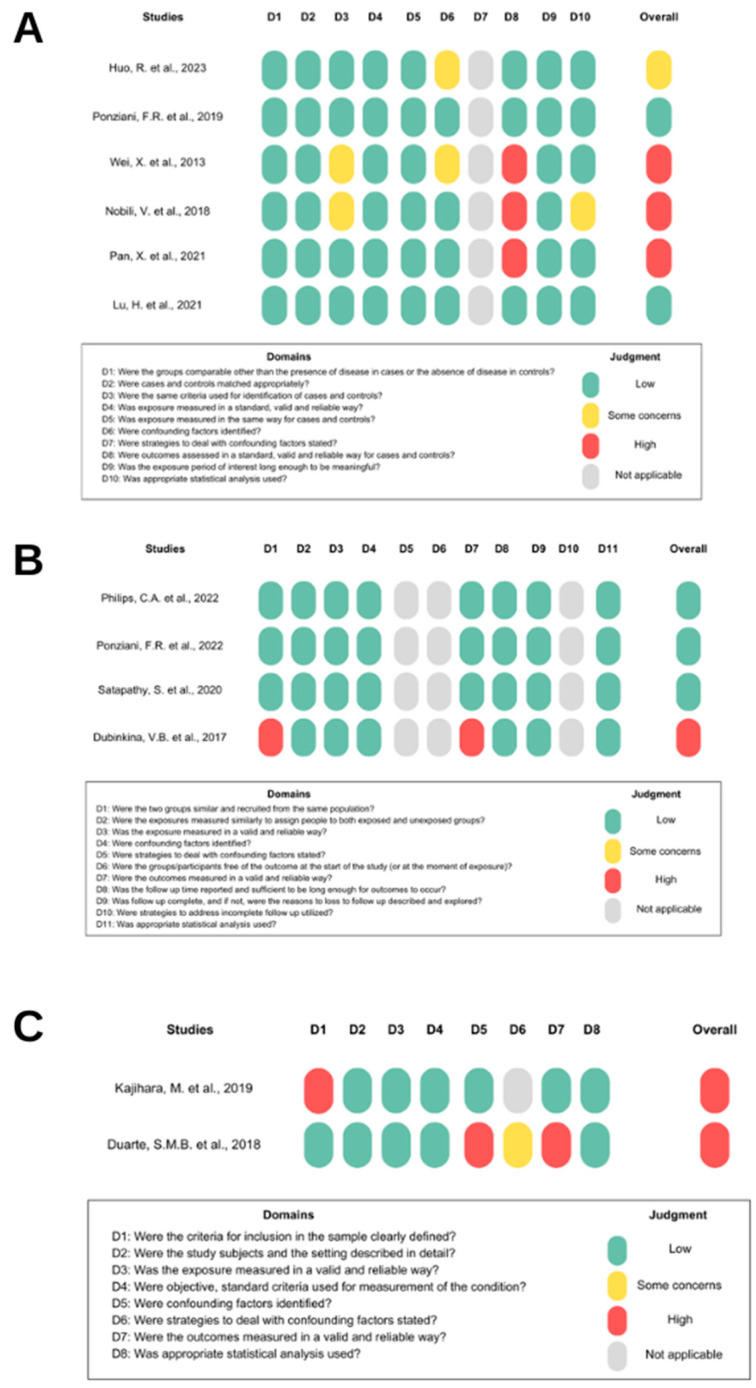
Risk of bias. (**A**) shows the risk of bias in case-control studies [4,19,28,30,34,35], (**B**) in cohort studies [20,33,34,43], and (**C**) in cross-sectional studies [18,29]. Domain “D” represents the items in the JBI checklist. “Overall” represents the study’s risk of bias based on judgment criteria.

**Figure 4 microorganisms-11-02999-f004:**
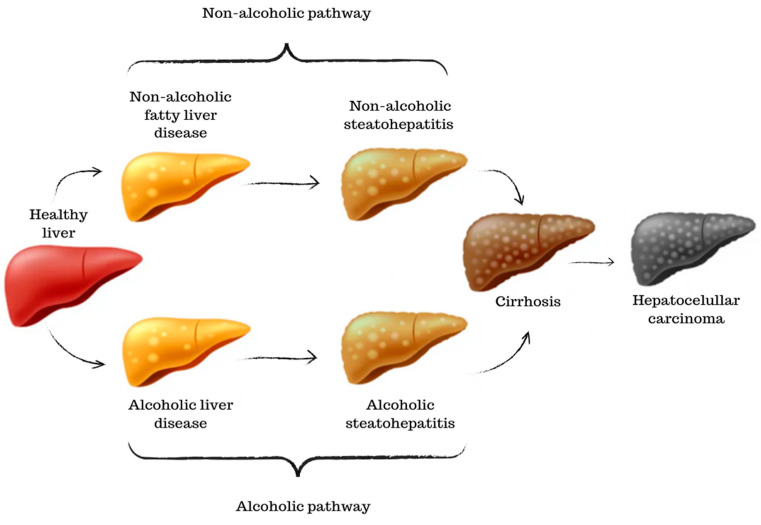
A graphical scheme of the progression and stages of liver diseases. Starting from a healthy liver until the development of hepatocellular carcinoma.

**Table 1 microorganisms-11-02999-t001:** Observational studies on *Bifidobacterium* levels in diseases.

Study	Country	Disease	Population	Type of Study	Type of Sample	N	Groups	Abundance in Disease	Clinical Relevance
Gu, Z. et al., 2020 [15]	China	ALD	Animal	Preclinical study	Cecum	64	7 Control	Down	Excessive alcohol alters gut bacteria, causing inflammation and other complications
							12 ALD		
							45 ALD + treatment		
Zhang, X. et al., 2021 [16]	China	HCC	Animal	Preclinical study	Stool	98	39 Control	Down	*Bifidobacterium* reduction linked to gut imbalance, chronic inflammation, immune dysfunction, and carcinoma development
							59 HCC		
Huo, R. et al., 2023 [4]	China	HCC	Human	Case-control	Stool	40	20 Control	Down	
							20 HCC		
Ponziani, F.R. et al., 2019 [17]	Italy	HCC	Human	Case-control	Stool	61	20 Control	Down	
							21 HCC		
							20 NAFLD + Cirrhosis		
Kajihara, M. et al., 2019 [18]	Japan	Cirrhosis	Human	Cross-sectional	Peripheral blood	80	14 Control	Down	Reduced *Bifidobacterium* levels in cirrhosis patients promote inflammation and worsen liver damage
							66 Cirrhosis		
Wei, X. et al., 2013 [19]	China	Cirrhosis	Human	Case-control	Stool	240	120 Control	Up	Unexpected result, but supported by related studies. No evidence of health issues with increased *Bifidobacterium*
							120 Cirrhosis		
Dubinkina, V.B. et al., 2017 [20]	Russia	Cirrhosis	Human	Cross-sectional	Stool	159	60 Control	Up	
							27 Cirrhosis		
							72 Alcohol dependence syndrome		
Juárez-Fernández, M. et al., 2021 [21]	Spain	NAFLD	Animal	Preclinical study	Stool	60	30 Control	Up	No evidence of harm from increased *Bifidobacterium* levels, but caution is advised due to similar trends in other diseases. Authors recommend caution when using *Bifidobacterium* as a probiotic
							30 NAFLD		
Zhang, Z. et al., 2021 [22]	China	NAFLD	Animal	Preclinical study	Stool	18	6 Control	Up	
							6 NAFLD		
							6 NAFLD + prebiotic		
Kang, K. et al., 2021 [23]	China	NAFLD	Animal	Preclinical study	Intestine	28	7 AFLD Control	Down	Imbalanced gut microbiota causes lower Bifidobacterium levels, increasing chronic inflammation and other clinical implications
							7 AFLD		
							7 NAFLD Control		
							7 NAFLD		
Liu, J.J. et al., 2023 [24]	China	NAFLD	Animal	Preclinical study	Stool	66	11 Control	Down	
							11 NAFLD		
							33 NAFLD + treatment		
							11 Fecal microbiota transplantation		
Li, W.F. et al., 2018 [25]	China	NAFLD	Animal	Preclinical study	Colon	32	8 Control	Down	
							8 NAFLD		
							16 NAFLD + treatment		
Zhang, B. et al., 2023 [26]	China	NAFLD	Animal	Preclinical study	Stool	60	10 Control	Down	
							10 NAFLD		
							40 NAFLD + treatment		
Ye, J.Z. et al., 2018 [27]	China	NASH	Animal	Preclinical study	Stool	24	12 Control	Down	Low levels of Bifidobacterium correlate with chronic inflammation and liver fibrosis
							12 NASH		
Nobili, V. et al., 2018 [28]	Italy	NASH	Human	Case-control	Stool	115	54 Control	Down	
							61 NASH		
Duarte, S.M.B. et al., 2018 [29]	Brazil	NASH	Human	Cross-sectional	Stool	23	10 Control	Down	
							13 NASH		
Pan, X. et al., 2021 [30]	China	NASH	Human	Case-control	Stool	75	25 Control	Down	
							25 NASH		
							25 NAFLD		

AFLD: Alcoholic fatty liver disease; ALD: Alcoholic liver disease; HCC: Hepatocellular carcinoma; NAFLD: Nonalcoholic fatty liver disease; NASH: Nonalcoholic steatohepatitis.

**Table 2 microorganisms-11-02999-t002:** Observational studies on *Bifidobacterium* levels after any treatment.

Study	Country	Disease	Population	Type of Study	Type of Sample	N	Groups	Treatment	Abundance by Treatment	Clinical Relevance
Warner, J. et al., 2021 [31]	USA	ALD	Animal	Preclinical study	Stool	36	6 Control	Human Beta Defensin 2 (hBD-2)	Up	Reduced of steatosis, hepatocellular death, and inflammation
							14 ALD			
							16 ALD + treatment			
Lin, Y. et al., 2021 [32]	China	ALD	Animal	Preclinical study	Stool	40	10 Control	Auricularia auricula Melanin (AMM)	Up	Harmful bacteria suppressed, gut barrier balanced, and reduced inflammation
							10 ALD			
							20 ALD + treatment			
Philips, C.A. et al., 2022 [33]	India	ALD	Human	Retrospective cohort	Stool	72	47 ALD + treatment 1	Fecal microbiota transplantation (1) and Pentoxifylline (2)	Up	Enhaced digestion and immunity through microbiota modulation
							25 ALD + treatment 2			
Ponziani, F.R. et al., 2022 [34]	Italy	HCC	Human	Prospective cohort	Stool	11	6 HCC responders	Tremelimumab and Durvalumab	Up	Improved gut microbiota promotes clinical remission of HCC
							5 HCC non-responders			
Lu, H. et al., 2021 [35]	China	Cirrhosis	Human	Case-control	Stool	53	29 Control	Lactitol	Up	Less endotoxin biosynthesis, lower inflammation, and disease severity
							24 Cirrhosis + treatment			
Zhang, Y. et al., 2018 [36]	China	NAFLD	Animal	Preclinical study	Stool	69	12 Control	Shenling baizhu powder	Up	Improved gut microbiota, reduced hepatic steatosis, and restored colonic mucosa enhance liver and intestinal health
							12 NAFLD			
							10 NAFLD + LPS			
							12 NAFLD + saline			
							11 NAFLD + treatment			
							12 NAFLD + probiotic			
Yang, F. et al., 2021 [37]	China	NAFLD	Animal	Preclinical study	Stool	60	10 Control	Fu instant tea	Up	Enhanced microbiota and short-chain fatty acids boost immune and digestive systems
							10 NAFLD			
							10 NAFLD + positive control			
							30 NAFLD + treatment			
Bao, T. et al., 2020 [38]	China	NAFLD	Animal	Preclinical study	Stool	60	15 Control	Inulin	Up	Improved gut barrier enhances nutrient absorption and health
							15 NAFLD			
							15 Control + treatment			
							15 NAFLD + treatment			
Yang, Z.D. et al., 2023 [39]	China	NAFLD	Animal	Preclinical study	Stool	18	6 Control	Inulin	Up	Increased short-chain fatty acids enhance the immune and digestive systems
							6 NAFLD			
							6 NAFLD + treatment			
Hu, W. et al., 2022 [40]	China	NAFLD	Animal	Preclinical study	Stool	21	7 Control	*Faecalibacterium prausnitzii* LC49/LB8	Up	Enhanced metabolism benefits patient health by improving NAFLD and metabolic function
							7 NAFLD			
							7 NAFLD + treatment			
Ghosh, S. et al., 2020 [41]	China	NAFLD	Animal	Preclinical study	Stool	30	10 Control	Inulin, fructooligosaccharide, and xylooligosaccharide	Up	Elevated short-chain fatty acids enhance immune function
							10 Unrestricted acess to treatment			
							10 Restricted acess to treatment			
Abernathy, B.E. et al., 2021 [42]	USA	NAFLD	Animal	Preclinical study	Cecum	72	12 Control	Polylactose	Up	Fat reduction, improved insulin sensitivity, and decreased systemic inflammation
							12 NAFLD			
							48 NAFLD + treatment			
Satapathy, S. et al., 2020 [43]	USA	NASH	Human	Prospective cohort	Stool	21	21 Pretransplant	Liver transplant	Up	Higher Bifidobacterium levels linked to less liver fat, protecting against NASH
							21 Posttransplant			
Bomhof, M.R. et al., 2019 [44]	Canada	NASH	Human	Randomized clinical trial	Stool	14	6 NASH + placebo	Oligofructose	Up	Liver steatosis, inflammation, fibrosis, and NASH activity have been significantly reduced
							8 NASH + treatment			

AFLD: Alcoholic fatty liver disease; ALD: Alcoholic liver disease; AMM: Auricularia auricula melanin; hBD-2: Human beta defensin 2; HCC: Hepatocellular carcinoma; LPS: Lipopolysaccharide; NAFLD: Nonalcoholic fatty liver disease; NASH: Nonalcoholic steatohepatitis.

## Data Availability

Not applicable.

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
