# Peer review of "The Role of Bifidobacterium in Liver Diseases: A Systematic Review of Next-Generation Sequencing Studies"

_microorganisms, 2023, doi:10.3390/microorganisms11122999_

Round 1
Reviewer 1 Report
Comments and Suggestions for Authors
General comment:
This is an interesting systematic review of 31 articles that showed reduced abundance of Bifidobacterium in hepatic pathologies. Bifidobacterium, a bacterial genus in the human gut microbiome, significantly influences liver diseases, a low levels of Bifidobacterium were associated with harmful biochemical and physiological parameters and adverse clinical outcomes.
Interventional studies using different treatments increased the abundance of the genus and improved clinical outcomes but the methods used to detect and measure the Bifidobacterium genus in the microbiome could affect the reliability of the research papers.
Comment 1: Introduction is short, it should be necessary to include some new updates citations about the gut-liver axis (Pezzino S, Sofia M, Faletra G, Mazzone C, Litrico G, La Greca G, Latteri S. Gut-Liver Axis and Non-Alcoholic Fatty Liver Disease: A Vicious Circle of Dysfunctions Orchestrated by the Gut Microbiome. Biology (Basel). 2022 Nov 6;11(11):1622. doi: 10.3390/biology11111622. PMID: 36358323; PMCID: PMC9687983).
Comment 2: Methods - A large number of studies could have been included, especially human studies.
Comment 3: 4.4. Bifidobacterium in NAFLD - This section needs to be expanded and more citations included, due to the important discrepancy in data about NAFLD.
Comments on the Quality of English LanguageWhile the manuscript is largely well-written, it will benefit from review by a native English speaker to tidy it up.
Author Response
This is an interesting systematic review of 31 articles that showed reduced abundance of Bifidobacterium in hepatic pathologies. Bifidobacterium, a bacterial genus in the human gut microbiome, significantly influences liver diseases, a low levels of Bifidobacterium were associated with harmful biochemical and physiological parameters and adverse clinical outcomes. Interventional studies using different treatments increased the abundance of the genus and improved clinical outcomes but the methods used to detect and measure the Bifidobacterium genus in the microbiome could affect the reliability of the research papers.
Reply: We thank the reviewer for the interesting suggestions, which allowed us to better highlight the focus of our study on NGS methods and related statistical analyses used to define the differential abundance of Bifidobacterium. Below we provide the point-by-point reply to them all. Changes in the revised version of the manuscript are highlighted in red.
Comment 1: Introduction is short, it should be necessary to include some new updates citations about the gut-liver axis (Pezzino S, Sofia M, Faletra G, Mazzone C, Litrico G, La Greca G, Latteri S. Gut-Liver Axis and Non-Alcoholic Fatty Liver Disease: A Vicious Circle of Dysfunctions Orchestrated by the Gut Microbiome. Biology (Basel). 2022 Nov 6;11(11):1622. doi: 10.3390/biology11111622. PMID: 36358323; PMCID: PMC9687983).
Reply: We have included additional information and updated citations. Please see lines 46-50 and 67-93.
Comment 2: Methods - A large number of studies could have been included, especially human studies.
Reply: Since our focus was on NGS methods, several studies were not included (i.e., studies that did not use NGS). Our aim with this focus on NGS methods was to provide, for the first time, an in-depth analysis of these methods in liver diseases and the identification of this particular genus. In addition, we already are exploring the use of Bifidobacterium as a probiotic or functional food in another study. Please note that using Bifidobacterium as a probiotic and identifying this genus in liver diseases are different aspects; that’s why we have chosen to write two different manuscripts. We have revised all sections of the manuscript to make our focus clearer to readers.
Comment 3: 4.4. Bifidobacterium in NAFLD - This section needs to be expanded and more citations included, due to the important discrepancy in data about NAFLD.
Reply: This section was expanded as suggested. Please see lines 355-375.
Reviewer 2 Report
Comments and Suggestions for Authors
This systematic review entitled “The Role of Bifidobacterium in Liver Diseases: A Systematic Review” explores the relationship between Bifidobacterium, a bacterial genus found in the gut, and liver diseases. The review analyzed 31 articles and found that Bifidobacterium levels are significantly altered in various liver diseases, and that this bacterial genus may play a role in the pathophysiology of these diseases. The review also suggests that Bifidobacterium may have therapeutic potential for managing liver diseases, although further research is needed to fully understand its mechanisms of action and potential clinical applications.
- Please revise the manuscript for the consistent correct use of Bifidobacterium
- The abbreviations in the tables should be defined in the table footer
- Line 218 – please remove 2
- Figure 4 is an original figure or is based on a similar figure? Please specify, also specify where was it produced
- The authors could mention also the administration of functional food that could positively impact the amount of Bifidobacterium in the microorganism (i.e. https://doi.org/10.1080/10408398.2022.2054934, https://doi.org/10.3390/biology11040553)
Overall, "The Role of Bifidobacterium in Liver Diseases: A Systematic Review" provides a comprehensive analysis of the relationship between Bifidobacterium and liver diseases. The study is well-structured and provides a detailed overview of the research methodology used to identify and analyze relevant articles. The findings of the study suggest that Bifidobacterium may play a significant role in the pathophysiology of liver diseases and may have therapeutic potential for managing these diseases.
However, there are some limitations to this study that could be improved in future research. For example, the study only analyzed a limited number of articles, and the sample size was relatively small. Additionally, the study did not provide a detailed analysis of the potential side effects or risks associated with using Bifidobacterium as a therapeutic intervention for liver diseases. Future research should aim to address these limitations and provide a more comprehensive understanding of the role of Bifidobacterium in liver diseases.
Author Response
This systematic review entitled “The Role of Bifidobacterium in Liver Diseases: A Systematic Review” explores the relationship between Bifidobacterium, a bacterial genus found in the gut, and liver diseases. The review analyzed 31 articles and found that Bifidobacterium levels are significantly altered in various liver diseases, and that this bacterial genus may play a role in the pathophysiology of these diseases. The review also suggests that Bifidobacterium may have therapeutic potential for managing liver diseases, although further research is needed to fully understand its mechanisms of action and potential clinical applications.
- Please revise the manuscript for the consistent correct use of Bifidobacterium
Reply: Done.
- The abbreviations in the tables should be defined in the table footer
Reply: Done.
- Line 218 – please remove 2
Reply: Done.
- Figure 4 is an original figure or is based on a similar figure? Please specify, also specify where was it produced.
Reply: This is an original figure drawn in PowerPoint.
- The authors could mention also the administration of functional food that could positively impact the amount of Bifidobacterium in the microorganism (i.e. https://doi.org/10.1080/10408398.2022.2054934, https://doi.org/10.3390/biology11040553)
Reply: We thank the reviewer for this suggestion, but we already are exploring the use of Bifidobacterium as a probiotic or functional food in another study. Please note that using Bifidobacterium as a probiotic and identifying this genus in liver diseases are different aspects; that’s why we have chosen to write two different manuscripts.
Overall, "The Role of Bifidobacterium in Liver Diseases: A Systematic Review" provides a comprehensive analysis of the relationship between Bifidobacterium and liver diseases. The study is well-structured and provides a detailed overview of the research methodology used to identify and analyze relevant articles. The findings of the study suggest that Bifidobacterium may play a significant role in the pathophysiology of liver diseases and may have therapeutic potential for managing these diseases. However, there are some limitations to this study that could be improved in future research. For example, the study only analyzed a limited number of articles, and the sample size was relatively small. Additionally, the study did not provide a detailed analysis of the potential side effects or risks associated with using Bifidobacterium as a therapeutic intervention for liver diseases. Future research should aim to address these limitations and provide a more comprehensive understanding of the role of Bifidobacterium in liver diseases.
Reply: We thank the reviewer for the interesting suggestions, which allowed us to better highlight the focus of our study on NGS methods and related statistical analyses used to define the differential abundance of Bifidobacterium. We have provided a point-by-point reply to them all. Changes in the revised version of the manuscript are highlighted in red. Since our focus was on NGS methods, several studies were not included (i.e., studies that did not use NGS). Our aim with this focus on NGS methods was to provide, for the first time, an in-depth analysis of these methods in liver diseases and the identification of this particular genus. We have revised all sections of the manuscript to make our focus clearer to readers. In addition, the potential side effects or risks associated with using Bifidobacterium as a therapeutic intervention for liver disease are explored in another study. Please note that using Bifidobacterium as a probiotic and identifying this genus in liver diseases are different aspects; that’s why we have chosen to write two different manuscripts.